# Improving RGB-Infrared Object Detection by Reducing Cross-Modality Redundancy

Qingwang Wang [1,2], Yongke Chi [1,2], Tao Shen [1,2,*], Jian Song [1], Zifeng Zhang [1,2] and Yan Zhu [2]

1 Yunnan Key Laboratory of Computer Technologies Application, Kunming University of Science and Technology, Kunming 650500, China; 14s005056@hit.edu.cn (Q.W.); chiyongke@stu.kust.edu.cn (Y.C.); songjian@kust.edu.cn (J.S.); zfzhang_9@stu.xidian.edu.cn (Z.Z.)
2 Faculty of Information Engineering and Automation, Kunming University of Science and Technology, Kunming 650500, China; zhuyan@kust.edu.cn
* Correspondence: shentao@kust.edu.cn; Tel.: +86-0871-65916593

**Abstract:** In the field of remote sensing image applications, RGB and infrared image object detection is an important technology. The object detection performance can be improved and the robustness of the algorithm will be enhanced by making full use of their complementary information. Existing RGB-infrared detection methods do not explicitly encourage RGB and infrared images to achieve effective multimodal learning. We find that when fusing RGB and infrared images, cross-modal redundant information weakens the degree of complementary information fusion. Inspired by this observation, we propose a redundant information suppression network (RISNet) which suppresses cross-modal redundant information and facilitates the fusion of RGB-Infrared complementary information. Specifically, we design a novel mutual information minimization module to reduce the redundancy between RGB appearance features and infrared radiation features, which enables the network to take full advantage of the complementary advantages of multimodality and improve the object detection performance. In addition, in view of the drawbacks of the current artificial classification of lighting conditions, such as the subjectivity of artificial classification and the lack of comprehensiveness (divided into day and night only), we propose a method based on histogram statistics to classify lighting conditions in more detail. Experimental results on two public RGB-infrared object detection datasets demonstrate the superiorities of our proposed method over the state-of-the-art approaches, especially under challenging conditions such as poor illumination, complex background, and low contrast.

**Keywords:** RGB-infrared images; object detection; multimodal feature fusion; mutual information minimization; illumination classification

## 1. Introduction

Object detection technology is one of the main technologies in the computer vision field. According to the different detection steps, the object detection technology can be divided into two types. One is the two-stage object detection technology. The detection process of this technology is to first generate a proposal and then perform fine-grained object detection. Its representative algorithms include Fast- RCNN [1], Faster-RCNN [2], etc. The other is the one-stage object detection technology. The detection process of this technology is to directly extract the image features in the network to predict the object category and location, and its representative algorithms include RetinaNet [3], SSD [4], etc.

Currently, object detection methods mainly use single-mode images related to the object detection task as training data. For example, Liu et al. used the RGB dataset collected in harsh environments to train models to detect vehicles and pedestrians [5]; Chen et al. used RGB images of KITTI and other datasets for cross-domain adaptive learning, making the model robust in different scenarios [6]. However, when the model using RGB images as training data is applied in complex environments, such as night, cloudy and rainy days, its detection effect is often unsatisfactory and its robustness is poor. Therefore, in view of the

robustness problem caused by using RGB as training data, some scholars have proposed the scheme of using the infrared image as training data. For example, Zhao et al. used the infrared image dataset to train the model for the task of pedestrian detection [7]; Dai et al. used the SIRST dataset to train the model for the task of infrared small target detection [8]. However, when the infrared image is used as the training data of the model, the detection accuracy of the algorithm is low since the infrared image lacks the texture information of the target. In addition, there are many small target problems in application scenarios such as remote sensing, and the detection performance of relevant algorithms needs to be improved, such as DLAO [9] and DRA [10]. It is worth mentioning that drawing on face detection technology is an effective way to improve the performance of related algorithms in detecting small targets [11].

In summary, only using single-modal images as the training data of the model will make it difficult to apply in the actual complex scene. Therefore, in order to overcome the above difficulties, many scholars have proposed the method of using multimodal image data as the training data for the detection model. Multimodal data have complementary advantages, such as RGB-infrared images. The advantage of infrared images is that they rely on heat imaging generated by the target and are not affected by external lighting conditions. The advantage of RGB images is that they can clearly capture the texture characteristics of the target, but are susceptible to lighting conditions. Therefore, research on multispectral object detection based on multimodal input has become a hot topic. In recent years, there's a growing joint use of RGB and infrared images for related research in various fields. For example, in the field of remote sensing, Dang et al. jointly use RGB and UAV near-infrared images to detect radish blight [12], and Y et al. jointly uses RGB and infrared images for terrain classification [13]. In the field of assisted driving, Li et al. jointly use RGB and infrared images for roadside Pedestrian detection [14], and Tian et al. jointly uses RGB and far-infrared images for 3D pedestrian detection in farmland [15]. However, even though multimodal data has great advantages, how efficiently fusing complementary information of multimodal data has become the core and difficulty of current algorithms.

Current cross-modal object detection algorithms based on RGB and infrared images mostly fuse complementary information by simply concatenating input features or designing an attention module to generate robust target features [16–23]. In terms of concatenating fusion, based on the two-stage object detection algorithm Faster-RCNN, Li et al. directly spliced RGB and infrared image features to obtain fusion features, and then input them into the RPN module [16]. Wagner et al. designed two direct fusion models to analyze their effects [17]. The first is early fusion, which directly combines RGB and infrared images at the pixel level. The second is late fusion, which fuses RGB and infrared image features in the final fully connected layer. Their analysis shows that the late fusion effect is more effective. In terms of attention-based mechanisms, Zhang et al. established two separate feature hierarchies for each modality, and then, by obtaining global features, the correlation between the two modalities was encoded into the attention module for subsequent fusion operations [18]. Zhang et al. proposed a cross-spectral fusion network, which added a channel attention module (CAM) [24] and a spatial attention module (SAM) [25] along the channel and spatial dimensions, making the network learn strongly meaningful features while suppressing interference at both channel and spatial levels [19]. Fu et al. proposed an adaptive spatial pixel-level feature fusion network. Through SAM and position attention module (PAM) [24] respectively, their network obtains the fusion weights of different spatial positions and pixels in the two feature maps, and uses the fusion weights to re-calibrate the original features of the RGB and infrared images to obtain a multi-scale fusion feature layer [20]. Cao et al. proposed a new multi-spectral channel feature fusion module to fuse the RGB and infrared image features, which uses the squeeze-and-excitation (SE) [26] module to obtain the fusion weights, and uses the fusion weights to recalibrate the original features [21]. However, these methods do not make full use of the complement information between different modalities, and do not give full play to the advantages of multimodal complementarity. In addition, these methods do not consider the redundant information

between multiple modalities when fusing complementary information, which leads to the fusion of redundant information when fusing complementary information.

The lighting condition of most RGB and infrared image datasets is artificially divided into two categories (day and night), such as the KAIST pedestrian dataset [27] and the DroneVehicle remote sensing dataset [28]. The criterion of classifying lighting conditions for the KAIST pedestrian dataset is that direct classification according to whether the shooting time is day or night. Similarly, the DroneVehicle remote sensing dataset divides the dataset into three lighting scenarios, daytime, nighttime, and dark nighttime, according to the time of the shooting. However, there are two obvious drawbacks in this way. The first is that the artificial classification is subjective, and it cannot accurately determine whether the light intensity of the image belongs to the day or night. For example, in the night scene category, there are also images with strong illumination caused by light sources such as street lights. The second is that only two lighting scenes are not comprehensive enough to fully reflect the actual lighting environment.

The above two problems restrict the further combined use of RGB and infrared images. Therefore, in order to solve the contradiction between complementary information and redundant information contained in different modalities, this paper proposes to design mutual information module to reduce redundant information between different modalities, and use an attention mechanism and illumination perception module to fuse complementary information of different modalities. In addition, given the subjectivity and incompleteness of artificial classification of lighting conditions, we propose a lighting condition classification method that uses histograms to statistics the gray values of RGB images and considers different gray value intervals as different lighting intensities.

In summary, the main contributions of this paper are listed as follows:

1.  This paper designs a mutual information module, which weakens the influence of redundant information on complementary information fusion.
2.  A lighting condition classification method based on histogram statistics is proposed to automatically classify lighting conditions in more detail, which facilitates the complementary fusion of infrared and RGB images.

The rest of this paper is structured as follows: In Section 2, we describe the network structure and methods in detail. Section 3 gives the details of our work and experimental results and related comparison to verify the effectiveness of our method. Finally, we summarize the research content in Section 4.

## 2. Methods

### 2.1. Overall Network Architecture

The overall architecture of the proposed RISNet is shown in Figure 1. RISNe is divided into four modules: feature extraction module, illumination perception module, feature fusion module, and detection module. Among them, the mutual information module proposed in this paper is embedded in the feature extraction module. The feature fusion module adds the CAM, and the detection module uses the light intensity information extracted by the illumination perception module, and combines the RGB and infrared image features to make the final detection.

Firstly, the input data to the network model is an RGB-infrared image pair. The feature map is obtained by extracting image features through the feature extraction module, and then the corresponding feature map is input into the mutual information module. Then the information entropy of the two feature maps is calculated, and the parameters of the feature extraction module are optimized by the information entropy, which reduces the redundant information of RGB-infrared image features. In the subsequent feature fusion process, the CAM captures the key regions of the RGB-infrared feature map, thus focusing on fusing the complementary information of the network attention region part. Based on the fact that the light intensity information is beneficial to improving the object detection effect, this paper adds the illumination perception module to extract the light intensity information from RGB images. Finally, the detection module combines the features obtained by the

feature fusion module and the light intensity information captured by the illumination perception module to predict the location and category of the object.

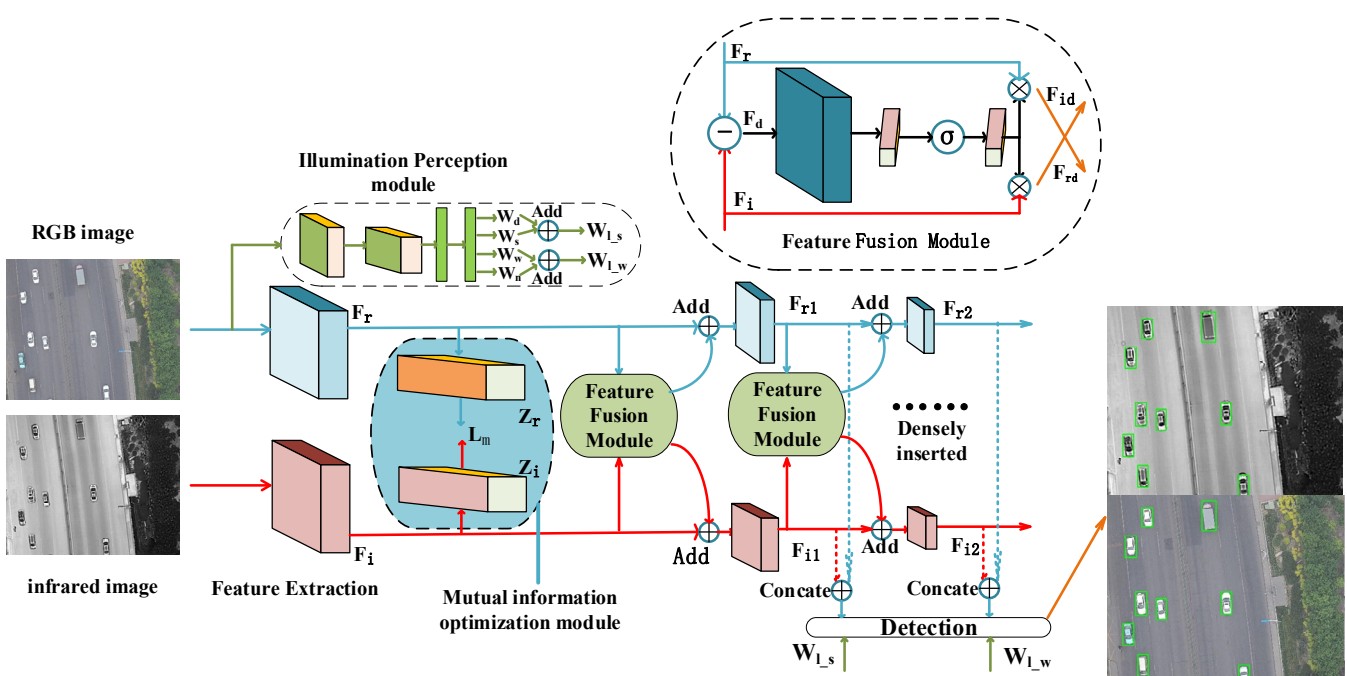

**Figure 1.** Overview framework of the proposed redundant information suppression network (RISNet). The RISNet consists of four parts: feature extraction stage, mutual information measurement module, feature fusion module and detection stage.

### 2.2. Mutual Information Module

One of the difficulties in target detection using RGB-infrared images is that there is both complementary and redundant information. Ideally, we want the network to fuse only the complementary features of RGB-infrared images. However, in practice, when the network fuses complementary information of RGB-infrared images, it inevitably fuses redundant information, resulting in poor detection performance of multimodal algorithms. Therefore, we filter the redundant information of RGB-infrared images in the feature extraction stage to resolve the contradiction between complementary information and redundant information in the feature fusion stage.

Therefore, based on the above ideas and inspired by the concept of information entropy in "information theory", we first treat RGB and infrared image features as two information distributions, and measure the amount of duplicate information between information distributions using mutual information [29,30]. Mutual information can be used to measure the amount of duplicate information between information distributions, which can optimize feature parameters, and finally reduce redundant information of infrared and RGB image features.

The mutual information module is shown in Figure 2. The input data of the module is the RGB-infrared image feature obtained by the feature extraction module. In order to reduce the parameters, computation of the network and facilitate the subsequent creation of the information distribution, we first map the RGB-infrared image feature to two low-dimensional feature vectors. In the module, we design two convolution layers and full connection layers to map the RGB-infrared image features to two $1 \times 6$ vectors $Z_r$ and $Z_i$. Then based on the two low-dimensional feature vectors $Z_r$ and $Z_i$, we create two information distributions (such as using $Z_r$ or $Z_i$ as the mean and $Z_i$ or $Z_r$ as the variance

to create a Gaussian distribution in this paper), and then calculate the mutual information of the two information distributions:

$$M(Z_r, Z_i) = H(Z_r) + H(Z_i) - H(Z_r, Z_i) \tag{1}$$

where $M(Z_r, Z_i)$ represents the mutual information of low-dimensional feature vectors $Z_r$ and $Z_i$. $H(Z_r)$ and $H(Z_i)$ indicate the information entropy of low-dimensional feature vectors $Z_r$ and $Z_i$, respectively. $H(Z_r, Z_i)$ represents the feature joint entropy of low-dimensional feature vectors $Z_r$ and $Z_i$. The relationship between information entropy, cross-entropy and relative entropy is as follows:

$$H(Z_i) = C(Z_r, Z_i) - D(Z_r||Z_i) \tag{2}$$

$$H(Z_r) = C(Z_i, Z_r) - D(Z_i||Z_r) \tag{3}$$

where $C(Z_r, Z_i)$ is the cross-entropy of low-dimensional feature vectors $Z_i$ and $Z_r$, $D(Z_r||Z_i)$ is the relative entropy of low-dimensional feature vectors $Z_r$ and $Z_i$. We then sum Equations (1)–(3), and obtain:

$$M(Z_r, Z_i) = C(Z_r, Z_i) + C(Z_i, Z_r) - D(Z_r||Z_i) - D(Z_i||Z_r) - H(Z_r, Z_i) \tag{4}$$

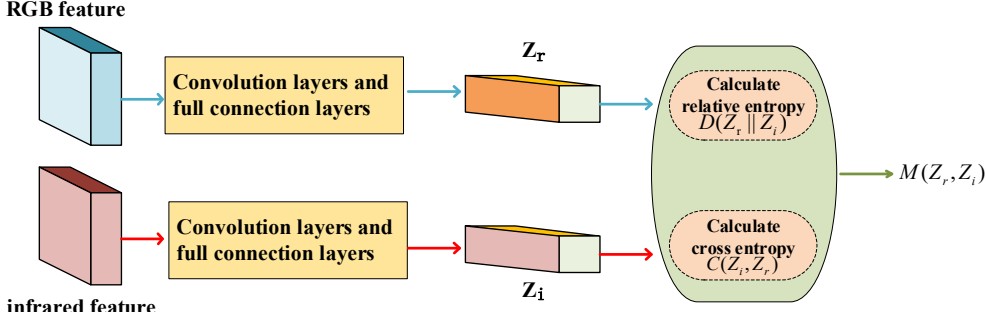

**Figure 2.** Mutual information module.

In order to simplify Equation (4) and consider the non-negative property of joint entropy, we delete $H(Z_r, Z_i)$, and finally get the formula for optimizing mutual information as Equation (5):

$$\mathcal{L}_m = C(Z_r, Z_i) + C(Z_i, Z_r) - D(Z_r||Z_i) - D(Z_i||Z_r) \tag{5}$$

$\mathcal{L}_m$ represents the amount of duplicate information of RGB-infrared image features. By optimizing $\mathcal{L}_m$, the feature extraction module parameters are optimized, and then the redundant information in RGB-infrared image features is filtered to obtain rich complementary information. Resolves the contradiction between the inability to distinguish and fusion complementary information and redundant information in the subsequent network fusion stage.

### 2.3. Lighting Condition Classification Based on Histogram Statistics

The subjectivity and incompleteness of artificially classified lighting conditions is another important reason that restricts the fusion of RGB-infrared images for object detection. Therefore, we propose a method based on histogram statistics to automatically classify lighting conditions in more detail. In order to further solve the incompleteness of classifying lighting scenes, we have made a detailed division of the gray value interval, covering as many lighting scenes as possible, to reflect lighting conditions in more detail.

First, we use the histogram to statistic the gray values of RGB images under different light intensities, and then divide the lighting scene according to the corresponding gray value intervals. As shown in Figure 3, we can clearly see from the histogram corresponding

to the RGB image that the gray value of the RGB image in full daytime is concentrated in (100, 255), the gray value of the RGB image at night but with strong light is concentrated in (34, 100), the gray value of the RGB image with weak light at night is concentrated in (10, 34), and the gray value of the completely dark RGB image is concentrated in (0, 10). Therefore, based on the above four gray value intervals, we divide the scene lighting condition into four categories: daytime, weak light at night, strong light at night and dark night.

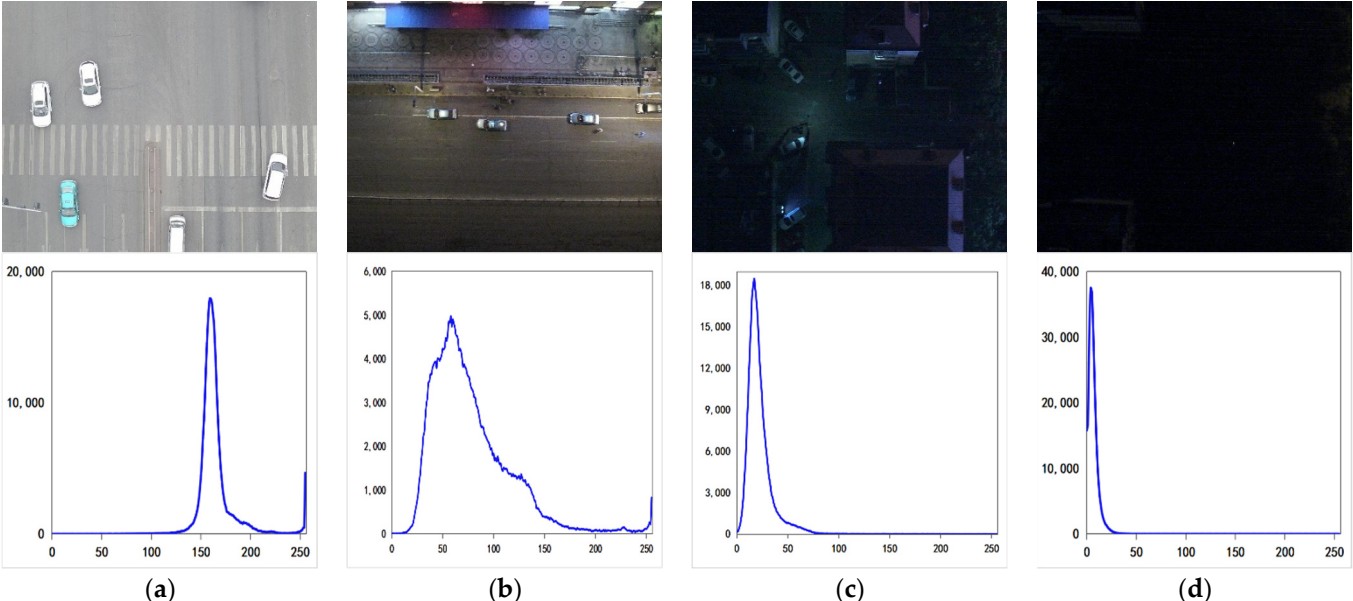

**Figure 3.** The first row is the RGB images of different lighting scenes, from (**a**–**d**) are daytime, weak light at night, strong light at night and dark night, and the second row is the corresponding histogram.

### 2.4. Illumination Perception Module

The illumination perception module is designed to extract the information of light intensity from RGB images, and to judge the reliability and importance of the RGB-infrared image features. If the RGB image has strong light intensity information, it means that the lighting condition is good. Visually, when the lighting conditions are good, the RGB image features can be used for detection. Otherwise, under poor lighting conditions, the infrared image features are mainly used to detect, taking advantage of infrared images.

As shown in Figure 1, the input data of the illumination perception module is the RGB image resized to $56 \times 56$. After the convolution layer and the full connection layer, four predicted values are generated: $W_d$, $W_s$, $W_w$, and $W_n$, which represent the confidence of the daytime, strong light at night, weak light at night, and dark night, respectively. Then, $W_d$ and $W_s$ are added, $W_w$ and $W_n$ are added, respectively. Then, pass the sigmoid function:

$$W_{l\_s} = \text{sigmoid}(W_d + W_s) \tag{6}$$

$$W_{l\_w} = \text{sigmoid}(W_w + W_n) \tag{7}$$

where $W_{l\_s}$ represents the weight of strong light intensity information and $W_{l\_w}$ represents the weight of light intensity information.

### 2.5. Feature Fusion Module and Detection Module

The core of the multimodal detection algorithm is how to effectively fuse the features between different modalities. Therefore, in order to effectively fuse the complementary information of RGB-infrared images, a channel attention mechanism is specially added to the feature fusion module, as shown in Figure 1. Firstly, the RGB-infrared image features are subtracted element by element, and the subtracted features represent the different features

of RGB infrared image features, which are regarded as complementary features of the RGB-infrared image. Then the different feature is input into the CAM to obtain the channel weight. Finally, the channel weight is multiplied by the corresponding RGB-infrared image feature, and added with another modal feature to obtain the final fused feature, as shown in the formula:

$$F_i' = CAM(F_r - F_i) \otimes F_r + F_i \tag{8}$$

$$F_r' = CAM(F_i - F_r) \otimes F_i + F_r \tag{9}$$

Among them, the $\otimes$ represents the multiplication operation, $F_r$ and $F_i$ are the feature of RGB and infrared image before fusion, $F_r'$ and $F_i'$ are the fused RGB and infrared image features.

The input data of the detection module is the light intensity information $W_{l\_s}$, $W_{l\_w}$ and the final fused feature $F_r'$ and $F_i'$, Through the additional illumination information, more robust detection can be achieved. A more detailed design of the feature fusion module and prediction module can refer to MBNet [31].

Therefore, the objective function of the network is divided into four parts: illumination loss $\mathcal{L}_i$, mutual information loss $\mathcal{L}_m$, classification loss $\mathcal{L}_{cls}$ and regression loss $\mathcal{L}_{reg}$, where the illumination loss $\mathcal{L}_i$ uses cross-entropy loss, the classification loss $\mathcal{L}_{cls}$ uses Focal loss, and the regression loss $\mathcal{L}_{reg}$ uses smooth $L_1$ loss. Our final objective function is:

$$\mathcal{L} = \lambda \mathcal{L}_m + \mathcal{L}_i + \mathcal{L}_{cls} + \mathcal{L}_{reg} \tag{10}$$

Among them, in order to balance the learning, the loss weight is set to 0.1.

## 3. Experiment and Analysis

In order to verify the effectiveness of the mutual information optimization module for reducing redundant information between modalities, and proves that the new lighting condition classification method is helpful to optimize the illumination perception module, we conduct experiments on the KAIST pedestrian dataset and DroneVehicle remote sensing dataset.

### 3.1. Dataset Introduction

KAIST pedestrian dataset contains 95,328 RGB-infrared image pairs, which contain three categories: person, people, and cyclist. Among them, the better-distinguished individuals are labeled as person. Not distinguishable individuals were labeled as people, and riding a two-wheeled vehicle were labeled as cyclists. Since the dataset is taken from consecutive frames of video, the adjacent pictures are not much different, so the dataset is cleaned to a certain extent, and pedestrians with only half or less than 50 pixels are eliminated. Therefore, the final training set contains 8963 pairs of images, of which 5622 were captured during the day, and 3341 were captured at night. The test set contains 2252 image pairs, of which 1455 were captured during the day and 797 were captured at night.

DroneVehicle remote sensing dataset contains 28,439 RGB-infrared image pairs, which are classified as vehicles and captured by UAVs equipped with cameras. The regional scenes are divided into urban roads, residential areas and highways. The lighting conditions during shooting are divided into dark night, night and day. The height of shooting is divided into 80 m, 100 m, and 120 m. The shooting angle is divided into 15°, 35°, and 45°. During the dataset calibration phase, RGB-infrared image pairs are cropped by affine transformation and region cropping to ensure that most of the cross-modality image pairs are aligned.

Some samples in the above dataset are shown in Figure 4.

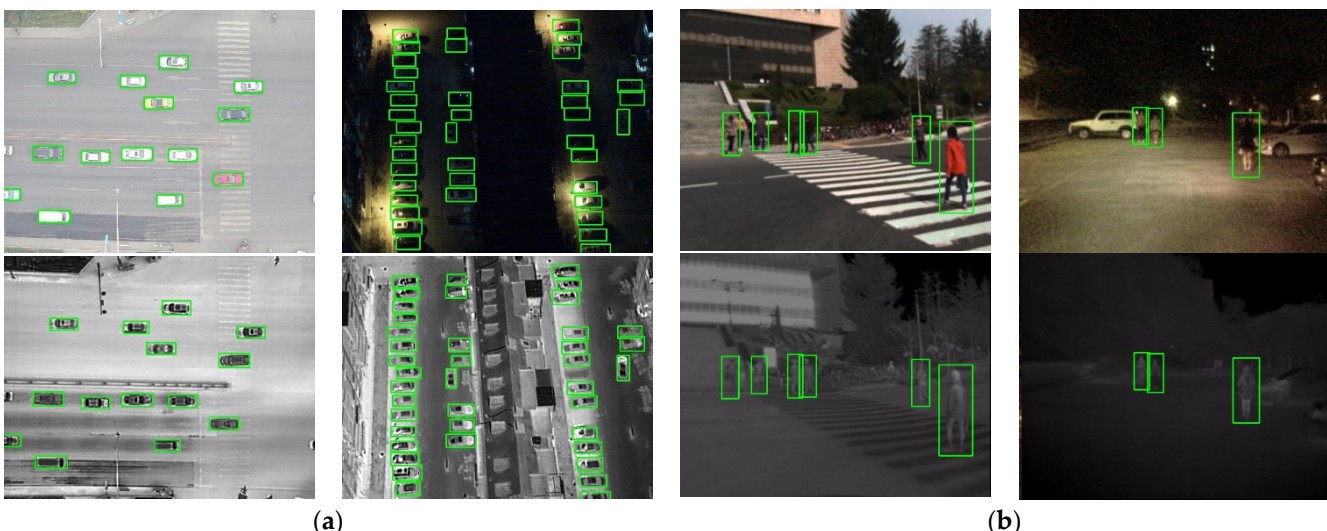

(**a**)                                                                            (**b**)

**Figure 4.** Sample visualization. (**a**) some images of DroneVehicle remote sensing dataset; (**b**) some images of KAIST pedestrian dataset. The first row is the RGB image, the second row is the corresponding infrared image.

### 3.2. Implementation Details

All experiments in this paper were implemented in Tensorflow on NVIDIA Tesla V100, and its operating system is ubuntu16.04, and the CUDA version is 10.2. For the processing of the dataset, this paper performs random color distortion and flip processing on the input image to achieve the purpose of increasing the diversity of data. For parameter settings during training, the whole network is trained by an adam optimizer for 30 epochs with an initial learning rate of 0.00001, and the learning rate is reduced by 10 times every 6 epochs. The batch size is set to 8. Followed by [32], the width of initial anchors are set to (25.84, 29.39), (33.81, 38.99), (44.47, 52.54), (65.80, 131.40) for four feature fusion stage with a single anchor ratio of 0.41. The dimension of feature vectors $Z_r$ and $Z_i$ is set to 6 through experimental analysis.

### 3.3. Evaluation Metrics

Precision, recall, average precision (AP), miss rate (MR), log-average miss rate (denoted as $MR^{-2}$) etc., are general performance evaluation indicators of object detection algorithms. In general, the better the classifier, the higher the AP and the lower the MR. In this paper, we adopt AP and $MR^{-2}$ as evaluation metrics. Among them, the calculation formula of precision and recall are as follows:

$$\text{Precision} = \frac{TP}{TP + FP} \tag{11}$$

$$\text{Recall} = \frac{TP}{TP + FN} \tag{12}$$

TP (true positive) represents the positive samples that were correctly detected; TN (true negative) represents the negative samples that were correctly detected; FP (false positive) represents the positive samples that were wrongly detected; FN (false negative) represents the negative samples that were wrongly detected. Among them, the standard for judging whether it is a positive or negative sample is IOU, and its calculation formula is as follows:

$$\text{IOU} = \frac{D \cap G}{D \cup G} \tag{13}$$

where D represents the predicted object bounding box area, and G represents the real object bounding box area. The IOU threshold in this paper is set to 0.5. If the IOU is greater than 0.5, the sample is considered to be a positive sample.

Taking Precision as the horizontal axis and Recall as the vertical axis, the area enclosed by the curve and the coordinate axis is the AP.

The MR reflects the ability of the model to correctly predict the purity of negative samples, and is calculated as follows:

$$MR = 1 - \text{Recall} \tag{14}$$

The calculation method of $MR^{-2}$: in the logarithmic coordinate system, evenly take 9 FPPI (False Positives Per Image) values from $10^{-2}$ to $10^0$, and average the 9 MR values corresponding to these 9 FPPI values. Among them, FPPI is calculated as follows:

$$FPPI = \frac{FP}{\text{the number of image}} \tag{15}$$

### 3.4. Analysis of Results

Firstly, comparison, and ablation experiments are carried out on the DroneVehicle remote sensing dataset, which proves that the method proposed in this paper is competitive. In order to further verify that the mutual information module is beneficial to RGB-infrared image complementary information fusion, a large number of comparative experiments were carried out on the KAIST pedestrian dataset, which verified that the mutual information module has a great effect on reducing redundant information between modalities.

#### 3.4.1. Experiments on the DroneVehicle Remote Sensing Dataset

We compare with DroneVehicle remote sensing dataset benchmarks [28], which include CMDet, UA-CMDet. We also use Yolov3 [33], Faster-RCNN [2] and MBNet [31] as the baselines to verify the effectiveness of the proposed method. Experiments show that the RISNet proposed in this paper has obvious advantages. The specific experimental results are shown in Table 1.

**Table 1.** Comparison results of different algorithms, the best AP indicators are generally black.

| Methods | Modality | AP |
|---|---|---|
| Yolov3 [33] | R | 56.40 |
| Faster-RCNN [2] | R | 44.59 |
| Yolov3 [33] | I | 59.32 |
| Faster-RCNN [2] | I | 54.55 |
| CMDet [28] | R + I | 62.58 |
| UA-CMDet [28] | R + I | 64.01 |
| MBNet [31] | R + I | 62.83 |
| RISNet(ours) | R + I | **66.40** |

Compared with the benchmark indicators of this dataset, the experimental results show that the AP value of our proposed RISNet is 66.40%, which has achieved the optimal effect, proving the effectiveness of RISNet for multimodal data detection.

In order to more clearly see the effectiveness of the mutual information module proposed in this paper, we show test image samples under different lighting scenarios, as shown in Figure 5.

From the comparative detection effect images of different lighting scenes in Figure 5, It is clearly seen that compared with the baseline, our proposed network is more robust, and still has a certain detection ability under various lighting conditions. In conclusion, it is proved that the mutual information module proposed in this paper and the histogram-based classification of lighting scenes are advanced, and provide a new idea for the joint utilization of RGB-infrared images.

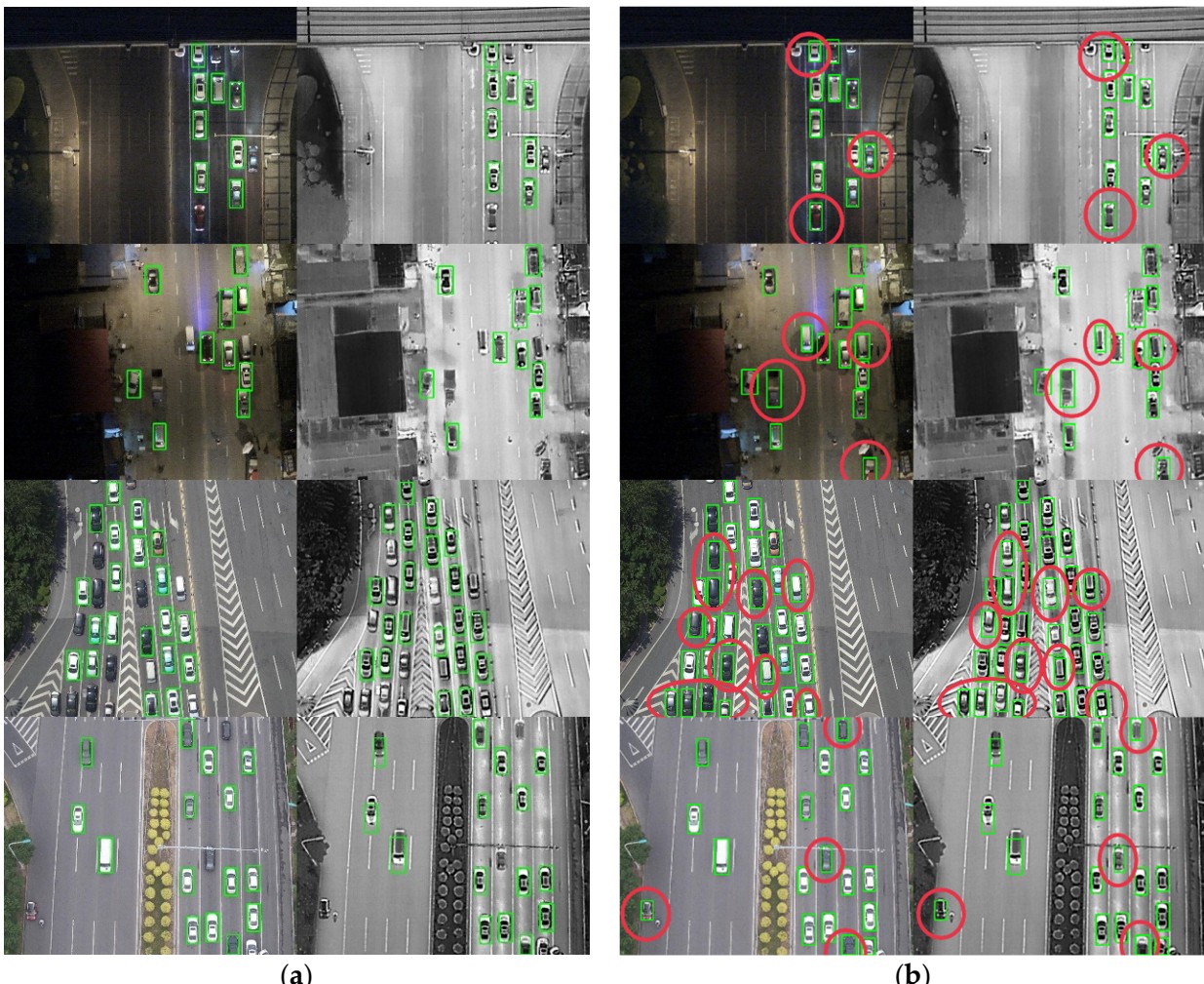

(**a**)                                            (**b**)

**Figure 5.** Visualization of baseline (**a**) and proposed RISNet (**b**) detection results on DroneVehicle remote sensing dataset. The first row shows the detection results in the daytime scenarios. Targets circled in red represent targets not detected by baseline but detected by RISNet.

### 3.4.2. Experiments on the KAIST Pedestrian Dataset

In order to further verify the effect of the mutual information module, we compare it with the current advanced multispectral pedestrian detection algorithms, such as MSDS-RCNN [32], AR-CNN [34], CIAN [18], MBNet [31]. We also use Yolov3 [33] as the baseline. The experimental results show that the proposed RISNet has obvious advantages over other algorithms. The experimental results are shown in Table 2.

**Table 2.** Evaluations on the KAIST dataset under all nine test subsets ($MR^{-2}$). The best $MR^{-2}$ indicators are generally black.

| Methods | Modality | All | Day | Night | Near | Medium | Far | None | Partial | Heavy |
|---|---|---|---|---|---|---|---|---|---|---|
| Yolov3 [33] | R | 52.85 | 43.25 | 72.88 | 33.94 | 61.28 | 91.43 | 68.55 | 81.58 | 87.39 |
| Yolov3 [33] | I | 32.67 | 37.62 | 23.38 | 6.29 | 43.59 | 78.04 | 52.15 | 62.91 | 78.72 |
| AR-CNN [34] | R + I | 9.34 | 9.94 | 8.38 | 0.00 | 16.08 | 69.00 | 31.40 | 38.63 | **55.73** |
| CIAN [18] | R + I | 14.12 | 14.77 | 11.13 | 3.71 | 19.04 | 55.82 | 30.31 | 41.57 | 62.48 |
| MSDS-RCNN [32] | R + I | 11.63 | 10.60 | 13.73 | 1.29 | 16.19 | 63.73 | 29.86 | 38.71 | 63.37 |
| MBNet [31] | R + I | 8.13 | 8.28 | 7.86 | 0.00 | 16.07 | 55.99 | 27.74 | 36.43 | 59.14 |
| RISNet (ours) | R + I | **7.89** | **7.61** | **7.08** | **0.00** | **14.01** | **52.67** | **25.23** | **34.25** | 56.14 |

Experimental results show that our proposed RISNet achieves the best $MR^{-2}$ in 8 subsets in the evaluation $MR^{-2}$ metric of nine subsets., Among, we obtain 7.89 $MR^{-2}$, 7.61 $MR^{-2}$, 7.08 $MR^{-2}$ in all day, daytime and night, respectively. In terms of pedestrian distance subsets, we get 0.00 $MR^{-2}$, 14.01 $MR^{-2}$, 52.67 $MR^{-2}$ on the near, medium, and far subsets, respectively. In terms of pedestrian occlusion degree subsets, we obtained 25.23 $MR^{-2}$, 34.25 $MR^{-2}$ on no occlusion, partial occlusion subsets, respectively. Obviously, the most $MR^{-2}$ value of our proposed RISNet is lower than the best competitive algorithm MBNet. The reason for this is that our proposed mutual information module can effectively reduce the redundant information of RGB-infrared features. For example, in dark conditions, the target is covered and the small target detection, the mutual information module reduces the redundant information of RGB-infrared features, so that the contour information of the target in the infrared image and the texture information of the target in the RGB image can fully play a role, and finally detected the target effectively.

In order to clearly compare the detection effects of different algorithms, we took out test image samples of different scenarios. As shown in Figure 6.

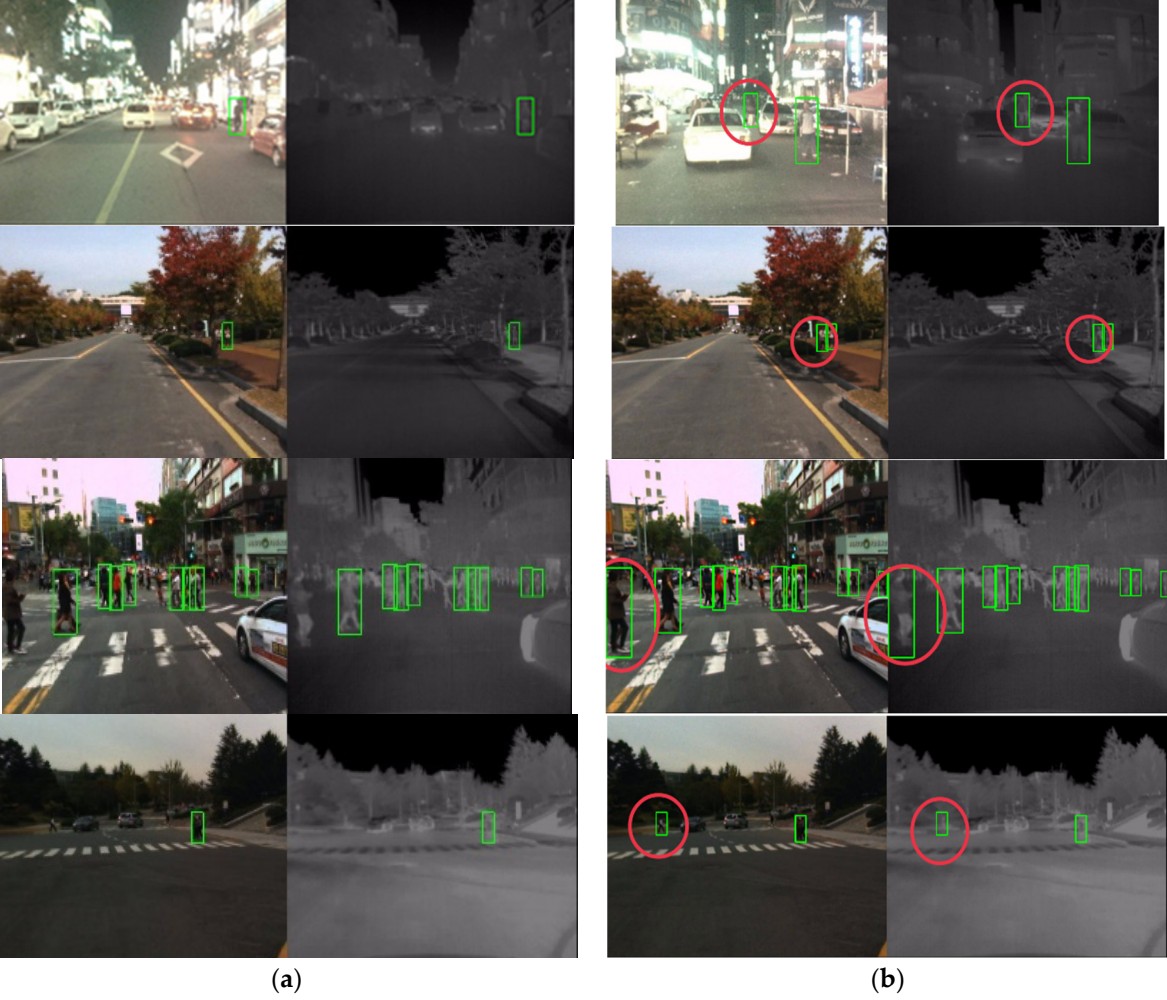

(**a**)            (**b**)

**Figure 6.** Visualization of MBNet (**a**) and the proposed RISNet (**b**) detection results on the KAIST dataset. The first row shows the detection results in the night scenarios. The second row shows the detection results in the daytime scenarios. The third row shows the detection results in the near scenarios. The fourth row shows the detection results in the far scenarios. Targets circled in red represent targets not detected by MBNet but detected by RISNet.

From the comparative detection effect images of different scenes in Figure 6, we can clearly see that MBNet is prone to miss detection in different scenes, but in the detection

effect images of RISNet, it can be accurately detected. Therefore, it is proved that the mutual information module proposed in this paper can improve the missed detection of the algorithm by reducing the redundant information of different modalities.

In order to compare the detection speed of various algorithms, we tested their FPS on different computing platforms and the results are shown in Table 3.

**Table 3.** FPS of different algorithms tested on different platforms. Ranking of computing performance (TFLOPs) of three computing platforms: GTX 1080 Ti (11.3 TFLOPs) > ITAN X (11 TFLOPs) > GTX 2060 (6.45 TFLOPs).

| Methods | Platform | FPS |
|---------|----------|-----|
| Yolov3 [33] | GTX 2060 | 63 |
| Yolov3 [33] | GTX 2060 | 63 |
| AR-CNN [34] | GTX 1080 Ti | 8 |
| CIAN [18] | GTX 1080 Ti | 14 |
| MSDS-RCNN [32] | TITAN X | 5 |
| MBNet [31] | GTX 1080 Ti | 14 |
| RISNet (ours) | GTX 2060 | 10 |

Comparing the FPS of each algorithm, it is obvious that the Yolov3 of the one-stage algorithm is the fastest, which effectively improves the detection speed at the expense of certain accuracy. Compared with the multimodal object detection algorithms, the fastest one is to reach 14 FPS, and the slowest one is only 5 FPS. Our proposed RISNet achieves 10 FPS, surpassing most multi-modal object detection algorithms, and can be more easily deployed to mobile terminals. In fact, FPS is not only related to the complexity of the model, but also related to the hardware platform tested. Since the computing performance of GTX 2060 is the worst, the proposed RISNet will achieve higher FPS on a better computing platform.

### 3.5. Ablation Experiments

In order to prove the effectiveness of the mutual information module and the new lighting condition classification method for the detection algorithm, we conducted ablation experiments in different lighting conditions and datasets, and the experimental results are shown in Tables 4 and 5.

**Table 4.** Ablation experiments on the DroneVehicle remote sensing dataset under different lighting conditions, where the baseline is MBNet [31], MI represents the mutual information module, and NLSC represents the new lighting condition classification method. The best AP indicators are generally black.

| Methods | Daytime | Strong Light at Night | Weak Light at Night | Dark Night |
|---------|---------|-----------------------|---------------------|------------|
| baseline | 70.46 | 70.60 | 52.41 | 43.98 |
| baseline + MI | 71.44 | **72.39** | **55.01** | 45.34 |
| baseline + NLSC | 70.81 | 71.05 | 54.67 | 45.00 |
| RISNet (ours) | **72.04** | 71.91 | 54.94 | **46.20** |

**Table 5.** Ablation experiments on the KAIST dataset under all nine test subsets, where the baseline is MBNet [31], MI represents the mutual information module, and NLSC represents the new lighting condition classification method. The best $MR^{-2}$ indicators are generally black.

| Methods | All | Day | Night | Near | Medium | Far | None | Partial | Heavy |
|---------|-----|-----|-------|------|--------|-----|------|---------|-------|
| baseline | 8.13 | 8.28 | 7.86 | 0.00 | 16.07 | 55.99 | 27.74 | 36.43 | 59.14 |
| baseline + MI | 9.02 | 9.90 | 7.28 | 0.00 | 14.82 | 53.12 | 25.89 | 35.07 | 60.93 |
| baseline + NLSC | 8.97 | 9.17 | 8.23 | 0.00 | 16.83 | 55.20 | 27.14 | 36.97 | 61.57 |
| RISNet (ours) | **7.89** | **7.61** | **7.08** | 0.00 | **14.01** | **52.67** | **25.23** | **34.25** | **56.14** |

As shown in Tables 4 and 5, when the mutual information module is added to the baseline model, the AP values of the four lighting scenes are improved by about 0.98% to 2.6%, which indicates that adding the mutual information module helps to eliminate redundant information of RGB-infrared image information and gives full play to the complementary advantages of RGB-infrared image information. But for different lighting scenarios, it cannot be improved to the same extent and equally. For example, lighting scenarios of weak light at night increased by 2.6%, while lighting scenarios of daytime only increased by 0.98%, which shows that the mutual information module can improve the algorithm detection performance, but it is still easily affected by the lighting scene, and cannot achieve the same degree and balance improvement of algorithm detection ability under different lighting conditions. When the new lighting scene classification method is introduced in the baseline model, the AP values of the four lighting scenes have been improved to the same extent and equally, which shows that the introduction of the new lighting scene classification method can enhance the robustness of the algorithm to different lighting conditions, but the improvement effect is not outstanding. On the basis of the baseline model, the mutual information module is added and the new lighting scene classification method is introduced, that is, RISNet proposed in this paper. The results show that under different lighting scenes, the improvement can be equal and excellent.

In conclusion, these results show that the mutual information module can improve the detection effect of the algorithm by reducing the redundant information of different modalities; it proves the scientificity of classifying lighting scenes based on the histogram, which makes the algorithm robust under different illumination conditions.

## 4. Conclusions and Discussion

In this work, we propose a new network named RISNet. It alleviates the problem of insufficient complementary information fusion caused by redundant information by designing the mutual information optimizer. Specifically, the mutual information module optimizes the feature extraction network parameters by calculating the mutual information of infrared and RGB image features, thereby reducing redundant information between modalities. Secondly, in view of the subjectivity and incompleteness problems faced by the current manual classification of lighting conditions, a method based on histogram statistics is proposed to solve the subjectivity of manual classification of lighting conditions, and through the detailed division of gray value interval, it covers as many lighting conditions as possible and solves the problem of the incompleteness of lighting conditions. The feature fusion module adds an attention mechanism to fully fuse the complementary information of infrared and RGB image features. In addition, an illumination perception module is added to the model to complement adaptive lighting perception features, making the detector robust to different lighting. From the experimental results, the proposed RISNet can reduce the redundant information and fully integrate the complementary information between multimodalities. In different lighting environments, it has better detection ability, reflecting the characteristics of high precision and strong robustness. Compared with the multimodal detection methods in recent years, our proposed RISNet demonstrates the superiority on most of the indicators.

There are a large number of small targets in remote sensing images, and the problem of small targets greatly affects the detection effect. Therefore, in future work, we will focus on how to improve the small target detection accuracy and further improve the detection effect of multimodal object detection methods. In addition, we will explore more effective ways to fuse multimodal features. Moreover, we continue to explore the relationship between complementary and redundant information of multimodal fusion in other computer vision tasks, such as RGB-infrared image semantic segmentation, RGB-Depth image semantic segmentation and so on.

**Author Contributions:** Conceptualization, Q.W. and T.S.; methodology, Q.W.; software, Y.C.; validation, Z.Z., J.S. and Y.Z.; formal analysis, Q.W. and Y.C.; investigation, J.S.; resources, T.S.; data curation, Y.C.; writing—original draft preparation, Y.C.; writing—review and editing, Q.W.; visualization, Y.C.; supervision, T.S. and Y.Z.; project administration, T.S.; funding acquisition, Q.W. All authors have read and agreed to the published version of the manuscript.

**Funding:** This research was supported by the Opening Foundation of Yunnan Key Laboratory of Computer Technologies Application (Grant No. 2021CTA08), the Yunnan Fundamental Research Projects (Grant No. 202101BE070001-008), the National Natural Science Foundation of China (Grant No. 61971208), the Yunnan Reserve Talents of Young and Middle-aged Academic and Technical Leaders (Grant No. 2019HB005), the Yunnan Young Top Talents of Ten Thousands Plan (Grant No. 201873), and the Major Science and Technology Projects in Yunnan Province (Grant No. 202002AB080001-8).

**Institutional Review Board Statement:** Not applicable.

**Informed Consent Statement:** Not applicable.

**Data Availability Statement:** The DroneVehicle remote sensing dataset is obtained from https://github.com/VisDrone/DroneVehicle, accessed on 29 December 2021. The KAIST pedestrian dataset is obtained from https://github.com/SoonminHwang/rgbt-ped-detection/tree/master/data, accessed on 12 November 2021.

**Conflicts of Interest:** The authors declare no conflict of interest.

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
