# Peer review of "Improving RGB-Infrared Object Detection by Reducing Cross-Modality Redundancy"

_remotesensing, doi:10.3390/rs14092020_

Round 1

Reviewer 1 Report

This paper proposes an Object Detection framework by Reducing Cross-Modality between RGB and Infrared. The idea is interesting, but there are a few major issues need to be fixed:

1- There are various parameters and hyper-parameters involved in this model, and it would be better to have a more in-depth discussion on how to tune them.

2- They should add a more in detailed section on ablation study.

3- Since in many cases the image has several small cars, they can borrow techniques developed in face detection for detecting many small faces in the image, into their framework. They can see their overview in this survey "Going deeper into face detection: A survey". 

4- They should provide comparison with more baseline object detection models.

5- Many of the relevant works on object detections, and small object detections are missing from the introduction, discussions, and references. They should add more relevant works. some are suggested below:

a- "Domain adaptive faster r-cnn for object detection in the wild." Proceedings of the IEEE conference on computer vision and pattern recognition. 2018.

b- "Going deeper into face detection: A survey." arXiv preprint arXiv:2103.14983 (2021).

Reviewer 2 Report

this paper proposes a redundant information suppression network (RISNet) which suppresses cross-modal redundant information and facilitates the fusion of RGB-Infrared complementary information.

Camera measuses RGB images, and infra-red sensor measures infrared images. The camera axis may not be identical to the axis of infra-red sensor.

How can you arrange two images whose axis are distinct from each other?

In other words, how do you handle the case where the position of camera is distinct from that of infra-red sensor?

"baseline+MI" and "baseline+NLSC" are examined in Table 2.

However, table 3 do not include "baseline+MI" and
"baseline+NLSC".  Can you implement these two algorithms in table 3?

Computational load (FramePerSecond) needs to be analyzed and compared to the-state-of-the-art in detail. 

Round 2

Reviewer 1 Report

This manuscript is improved and I think it is in a better shape to be published. Thanks to the authors for addressing many of my concerns.

Reviewer 2 Report

I am satisfied with revision. Thank you.

This manuscript is a resubmission of an earlier submission. The following is a list of the peer review reports and author responses from that submission.